# Barriers to Learning Healthcare-Associated Infections Prevention and Control during Clinical Practicum among Nursing Students in Korea: A Focus Group Study

**DOI:** 10.3390/ijerph20146430

**Published:** 2023-07-22

**Authors:** Eunyoung Park, Hyung-Ran Park, Ji-Hye Lee

**Affiliations:** 1College of Nursing, Chungnam National University, Daejeon 35015, Republic of Korea; eypark@cnu.ac.kr; 2Department of Nursing Science, College of Medicine, Chungbuk National University, Cheongju 28644, Republic of Korea; olive1512@naver.com

**Keywords:** healthcare-associated infections, nursing students, barriers, qualitative research

## Abstract

Healthcare-associated infections (HAI) refer to infections that patients may acquire from healthcare facilities through nursing activities. Nursing students involved in patient care are regularly exposed to an array of pathogens and clinical practicum is essential for them to appropriately practice HAI prevention and control. This study aimed to explore the barriers to learning HAI prevention and control experienced by nursing students during their clinical practicum. A qualitative study was performed using focus group interviews. A total of 12 nursing students from South Korea, consisting of six third-year students and six fourth-year students, were enrolled. Both groups had taken clinical practicum courses. Data were collected using semi-structured questions and analyzed with conventional content analysis. Barriers experienced by the participants when learning HAI prevention and control were limited learning opportunities, inadequate infection control-related knowledge, inadequate practicum experience, and passive learning attitudes. Addressing the identified barriers would allow nursing students to effectively acquire infection control competencies during their clinical practicum.

## 1. Introduction

Healthcare-associated infections (HAI) refer to infections that patients acquire while receiving treatment and nursing care in a healthcare environment. HAI not only increase morbidity and mortality rates in the healthcare setting but also have serious adverse effects on healthcare services by extending patients’ hospital stay and increasing their healthcare costs [1,2]. Pathogens that cause HAI are mostly found on hands and HAI can be largely prevented through hand hygiene, use of personal protective equipment (PPE), safe injection practices, safe disposal of sharps, use of disinfected equipment, compliance with evidence-based guidelines for maintenance of a clean environment, and infection surveillance and feedback [3,4]. During the coronavirus disease 2019 (COVID-19) pandemic, hand hygiene, use of PPE, and rigorous environmental management were found to lower the incidence of *clostridioides difficile* infections (CDI) [2,5], rates of central line-associated bloodstream infections (CLABSI), catheter-associated urinary tract infections (CAUTI), ventilator-associated infections, and methicillin-resistant *Staphylococcus aureus* (MRSA) infections in hospitals, thereby highlighting the importance of HAI control in hospital environments [2,5,6].

The Korean government established a nationwide network, called the Korean National Healthcare-Associated Infections Surveillance System in 2006 and has conducted prospective surveillance to ensure strict compliance with infection prevention and control practices by all healthcare professionals involved in patient care and treatment in hospital settings [7,8,9]. Prevention and control of HAI require all healthcare professionals, including nursing students who are involved in healthcare, to be well-aware of multifaceted infections [3,4,7]. As members of the healthcare team closely involved in providing nursing care to patients, nursing students play a critical role in preventing the spread of pathogens and infectious diseases during their clinical practicum [3]. The World Health Organization recommended incorporating standardized education for HAI prevention and control in the nursing school curriculum for students participating in patient care [3]. Korea’s nursing school curriculum comprises 1000 or more clinical hours; yet it does not include specialized courses or recommended programs for standardized HAI prevention and control [10,11]. As a result, students are not adequately educated about HAI prevention and control practices in clinical settings [3]. Particularly during the COVID-19 pandemic, exploring the challenges experienced by nursing students in learning HAI prevention and control in clinical practicum would be significant, as its prevention and control are considered an essential characteristic of healthcare professionals, and its importance is consistently emphasized, even with the dwindling pandemic.

Past qualitative studies on HAI prevention and control among nursing students explored their overall experiences in patient care in COVID-19 care units [12,13,14,15,16,17,18,19] or their experiences in adapting to clinical practicum during the COVID-19 pandemic [20,21]. As these studies present only the general experiences of nursing students in the unique COVID-19 situation, it is necessary to investigate the barriers to learning HAI prevention and control during clinical practicum to propose the direction of education on this topic.

### Research Question

This study aimed to explore the barriers to learning HAI prevention and control among nursing students; its specific study question was “What were the barriers to learning HAI prevention and control during your clinical practicum?”

## 2. Materials and Methods

### 2.1. Design

In this study, we used a qualitative research method with focus group interviews (FGI) to explore the barriers to learning HAI prevention and control among nursing students. This is a more effective methodology than individual interviews for examining participants’ opinions that are dynamically shaped as they interact [22]. We adhered to the consolidated criteria for reporting qualitative research (COREQ) checklist [23].

### 2.2. Participants and Setting

Third and fourth-year nursing students in Korea, with clinical practicum experience in a hospital setting, voluntarily participated in this study. The most desirable size of focus groups was four to eight participants [22]. We, therefore, purposively sampled our participants to select the most desirable study sample for studying our phenomenon of interest. We selected two groups—one group of third-year nursing students and one of fourth-year students—each consisting of six students. The participants completed five practicum courses (10 credits) per year for a total of 450 clinical hours. Third-year students completed practicums in internal medicine, surgery, pediatrics, maternity, operating room, and emergency room, while fourth-year students completed practicums in mental health, internal medicine, surgery, geriatrics, and intensive care units. The site for the clinical practicums was a national university general hospital in Cheongju, Korea. This hospital was designated for isolation beds during the COVID-19 pandemic. However, the nursing students were not allowed inside the isolation ward during their clinical practicum.

### 2.3. Data Collection

A focus group study was conducted in a quiet seminar room at the school from December 2019 to October 2020. The interviews were facilitated by a moderator with qualitative study experience and the interviewer continuously noted group dynamics during the interviews. Two research assistants attended the interviews to take notes of non-verbal expressions and interactions. Each interview lasted 90 min, and a total of three interviews for third-year students and two interviews for fourth-year students were conducted until data saturation. The interviews were recorded as MP3 files and later transcribed by research assistants and reviewed by the researchers. Data were collected using a semi-structured questionnaire developed based on a literature review [10,24]. The key interview question was “What were the barriers to learning HAI prevention and control during your clinical practicum?” Other questions used during the interview are shown in Table 1.

### 2.4. Analysis

To describe the barriers to learning HAI prevention and control during clinical practicum, we performed conventional content analysis, a type of inductive analysis [25,26]. In the first phase, the interview recordings were transcribed verbatim and read over, while cross-checking with the recordings. To identify meaningful words, phrases, and paragraphs about experiences in which the learning of HAI prevention and control was hindered, the transcripts were read repeatedly. We focused on latent meanings in the context to grasp a general idea about the barriers described by the participants.

In the second phase, the statements were coded, categorized, and abstracted. For coding, meaningful statements that suggested barriers in learning HAI control were underlined and the codes were named in the empty space on the transcripts. After coding the statements, similar codes were clustered into the same sub-category. The codes and sub-categories were compared again against the raw data to check for any redundant or unassociated content and were modified accordingly. For abstraction, similar sub-categories were clustered into overarching categories. The researchers collaborated to review one another’s opinions during the abstraction stage and agreed on the final categories derived. Each process of this study was performed in Korean. For publication, this manuscript was translated into English by a professional translator and edited by an expert.

To ensure rigor of the study, trustworthiness criteria were used [27]. Credibility was ensured by confirming with the participants that the derived categories and sub-categories accurately represented their experiences. Dependability was ensured by recording all the interviews and taking field notes on nonverbal communications during the interviews. Fittingness was ensured by collecting in-depth data until saturation and precisely documenting the results using quotes that represented the context well. Confirmability was ensured by collecting data using blanketing and having two researchers independently analyze the codes and sub-categories before deriving the categories.

### 2.5. Ethical Considerations

This study was conducted after obtaining approval from the Institutional Review Board of the corresponding author’s affiliation (CBNU-201911-BMSB-028). Considering that the study population comprised students, a researcher with a PhD and qualitative research experience, unrelated to evaluation of the students’ courses, was set as the moderator to provide an explanation about the purpose and procedure of the study and facilitate the interview. Prior to beginning the interview, the participants were provided detailed written information about the purpose, procedure, freedom to withdraw from the study at any time, as well as confidentiality and anonymity. The participants voluntarily signed the informed consent form to participate in the study.

## 3. Results

Two groups of six students were enrolled. There were 2 male and 10 female students, and their age ranged from 21–25 years (Table 2).

Four themes and eight categories were identified, as shown in Table 3.

### 3.1. Limited Learning Opportunities

Participants experienced limitations in learning about HAI prevention and control during their clinical practicum because nursing theory and practicum courses primarily focus on diseases. In particular, regardless of the pandemic, the hospital units often restricted students from participating in nursing activities for patients with HAI, and students were sometimes prevented from actively participating in HAI prevention and control activities either by nurses or patients.

#### 3.1.1. Not Many Learning Opportunities

As nursing theory courses take disease-centered approaches, the participants were only able to learn about HAI prevention and control in some courses, such as microbiology and fundamentals of nursing offered in the first and second years of study. Particularly, as most assignments given in practicum courses were case studies that apply the nursing process, the students tried to choose patients they could interact with freely to collect enough data for their assignments; this further reduced their opportunity to learn about HAI prevention and control for patients with infections.
*You know, in our classes, we primarily learned about diseases… because we focused so much on diseases during our studies, I forgot everything about infection control, like standard precautions, and by the time I went to clinicals, I just kept saying to myself, “Oh right, we learned about this.”*(Participant 5)
*I think seeing the patient in person helps to learn more about the particular disease. So, I think that is why I could not learn more about infection control… The clinicals are usually structured as case studies, and because it involves talking to the patients to get more subjective data, for their case studies, everybody tends to choose patients, whom they encounter as much as possible. So, I think that is why I could not choose patients with an infection, for my case study.*(Participant 12)

There is only one infectious diseases unit at the clinical site, so only some students were given an opportunity to learn about HAI control in-depth, even before the COVID-19 pandemic. Furthermore, nursing students were generally not allowed to participate in invasive nursing activities. The students stated that they had few opportunities to experience standard precaution-related activities such as safe injection practices.
*I had my rotation in an infectious diseases unit so I was able to see a lot of cases of patients with infections, but you know not everyone gets to do their rotations at such units. So, I think not everyone has such opportunities to learn about preventing… such infections during their clinicals.*(Participant 4)

#### 3.1.2. Limited Face-to-Face Interactions with Patients with an Infection

The participants experienced that their preceptors restricted them from entering the rooms of patients with isolation precautions. The students had rotations in units with patients with infections such as MRSA, CDI, scabies, tuberculosis, and human immunodeficiency virus (HIV), but they mentioned that their preceptors prevented them from meeting these patients as much as possible. In addition, students were not encouraged to participate in infection control activities for patients with infections, even in patient rooms without isolation precautions. During the COVID-19 pandemic, students were also limited in their practical learning experiences with patients with infections, as they were not allowed to enter COVID-19 units.
*In the case of COVID-19, patients come up to the general med-surg floors where we are placed only after their test is negative… the hospital tells us not to go into rooms of patients with contact precautions, the single-person rooms… We had no chance of wearing N95 masks or AP gowns and I have never done it before… So, even after COVID started, we did not really do much during our clinicals. Students have to wear a face mask and goggles during clinicals; so, we just wore those.*(Participant 3)
*When I was placed in an infectious diseases unit, there were patients with MRSA… and VRE… but I was told not to go into those rooms at all… So, I just knew that we had those patients and I did not go into those rooms… The hospital excludes… I think, they kept students away from patients with airborne or droplet precautions, as far as possible.*(Participant 8)

Students who had restricted in-person encounters with patients with infections stated, that they had no rememberable experiences about learning HAI control because they were not able to do anything to participate in HAI control.
*Even if we learn it at school or they talk about it in the unit and our preceptors teach us about it, we cannot actually see these patients; so, it just flies by us… so, I think nothing was really rememberable.*(Participant 5)

#### 3.1.3. Inability to Be Proactive in Clinical Practicum Settings

Regardless of the pandemic status, the students were unable to proactively participate in nursing activities for patients in isolation rooms or with infections to learn about HAI prevention and control. The students were discouraged by the nurses’ evaluation that they were likely to be the vectors of pathogen transmission owing to their inadequate HAI prevention and control competencies. Moreover, students mentioned that inappropriate practices demonstrated by their preceptors hindered their own practice.
*Honestly, I think the preceptors did not really like us coming in at all… because we might make things worse… what they always said is, that students do not sufficiently practice hand hygiene and they blamed the infections on students… so, I was very careful about being proactive to do anything.*(Participant 11)
*You know, ever since COVID-19, wearing a face mask and other PPE has become a sensitive issue… However, there were a lot of patients using contact precautions at my rotation site. Then, I had to wear gloves too. There are gloves in every room, but I saw that a lot of the preceptors did not wear gloves when providing patient care, especially when they quickly do things and stuff… but when I wear gloves to take vitals, the caregivers kind of give us this look. At first, I really did try to put gloves on… but they kept giving us weird looks; so, I just stopped wearing gloves when taking vitals. That was a bit of a challenge.*(Participant 1)
*I went to take vital signs, and you know the sign is up there on the bed, “contact precautions.” After I saw that, I was going to don the gown, but the patient told me that the nurses do not wear it, and only the students try to wear it. So, I was kind of embarrassed and did not wear it, although I know that it is to be worn not only to protect ourselves, but also other patients. However, the patients do not like that. And, that was kind of hard… The patients hated it so much; so, we were debating whether to do it and decided not to… We just took care of that patient last and washed our hands really thoroughly after that.*(Participant 2)

Students who were assigned to a nurse preceptor, who would not inform or educate them about patients with infections when they first began their hospital clinical rotations, faced challenges in autonomically performing HAI control because they were not aware which patients had infections.
*At first, there was a time when I did not know, but found out later, during the handoff, that a patient had a communicable disease … My reaction was like “Uh oh,” when I heard about it. I was so concerned and thought to myself, why did she not tell me? … I just touched that patient without any protection… She did not also tell us about other patients with contact precautions; so, we just looked through the electronic medical records.*(Participant 5)

Furthermore, the participants were concerned that their use of the PPE provided by the hospital would deplete the supply available for the hospital’s staff. Such concerns about possible supply shortage posed restrictions to their PPE usage and participation in caring for patients with infections. Hesitancy regarding the use of supplies had been identified as a barrier to learning among nursing students, even prior to the massive supply shortage brought upon by the COVID-19 pandemic.
*Even before COVID, it was a bit hard to go in… I just stood outside the room, even during the rounds… If we go in without the preceptors, then too, we have to use supplies, and we were concerned about whether it is okay for us to use them. We have to use the gowns or face masks and stuff available in front of the rooms, and as these are disposable, we have to throw them away after use. But we are students; so, we were thinking whether it was okay for us to keep using them. Would it not affect the supply?*(Participant 3)

### 3.2. Lack of Knowledge about Infection Control

The participants experienced that they had inadequate knowledge about the details of each type of precaution, types of infectious pathogens, and routes of transmission when they attempted to provide interventions appropriate to the type of precaution for a particular patient with infection during their rotations. In particular, their memories of what they had previously learned were lost, which further hindered their practice.

#### 3.2.1. Lack of Specific Knowledge about Transmission Routes

The participants were aware of precautions—standard, contact, and airborne—for infection control, that they had learned in theory courses, but they had inadequate knowledge about the compositions of each type of precaution and the required activities appropriate for each precaution. Moreover, the students stated that they required precise and accurate knowledge about the definition and composition of each precaution, as opposed to learning new knowledge about HAI prevention and control, even after the outbreak of COVID-19.
*Even if it is an emerging infectious disease… it is something we have been learning; so, instead of something new… it wasn’t like I knew that content (or had already learned it before)… in depth or extensively, but I just had shallow knowledge about it, like hand hygiene and use of PPE… I just had a broad sense of knowledge like that… I did not know that in detail… like what is required for each, and what makes it up. I did not know; so, I could not do anything because I did not know what to observe.*(Participant 2)
*You know, standard precautions are basic. Even during the pandemic, such basic knowledge is what I had already learned before; so, it did not feel like something new. However, I needed to know these well… like know exactly what the standard precautions are… what are the things on standard precautions… I just knew about hand hygiene and PPE, and did not know the rest.*(Participant 6)

In particular, the participants lacked adequate knowledge about the pathogens that fall under each type of precaution and their pathophysiology, such as route of transmission which presented challenges for them in learning about HAI prevention and control practices.
*There was a patient suspected of measles in our unit, and at that time, I did not exactly know that measles involves airborne precautions… I had limitations about what diseases take each route of transmission.*(Participant 11)

#### 3.2.2. Fading Memories of Learned Knowledge over Time

The participants mentioned that as they learn about infection control in their first and second years of nursing school, and their clinical rotations begin in the third year, they practically lost anything previously learned which posed challenges in their practice during clinical practicum.
*I learned theory in my second year, and it was taught when I had rotations in surgery during the first semester… the head nurse in the hospital once taught us… but I could not remember it that well during clinicals in the following semester. I think it did not come to me quickly because I had not been hands on for a long time since then and I had never tried that after learning about it.*(Participant 2)

### 3.3. Inadequate Training Experience

The participants had difficulty performing hand hygiene and donning PPE when they first began their clinical rotations at the hospital because they were not proficient in skills. Particularly, they floundered when they actually faced with situations calling for infection prevention in clinical settings and thus could not effectively perform HAI control practices. Hence, the students stated that they should learn infection control based on actual scenarios.

#### 3.3.1. Inadequate Infection Management Skills

The participants were too nervous to recognize HAI control practices when they first began their clinical rotations at the hospital and some mentioned that they merely pretended to don the PPEs because they were not familiar with the products used in the hospital. They mentioned that applying their knowledge is hindered if they do not have repeated and adequate hands-on practice beyond simply acquiring knowledge.
*I think I kept forgetting at first. I have to perform hand hygiene after touching every patient, but I wasn’t aware of that, and I was nervous, and it was my first rotation… Sometimes, I just forgot performing hand hygiene and remembered later… and it was my first time wearing an N95 mask during my internal medicine rotation. When wearing the mask, it should cover my entire face, but I felt like the mask was not fitted well, and there were empty spaces and stuff.*(Participant 4)
*We learned about it, but when it was actually time to use it, I could not remember what I had learned… I think I did not perform it correctly because of nervousness and stuff…I had practiced it a couple of times when we had learned about it… but even if I knew it, I could not really apply it during my rotations… it does not come to mind quickly… I think I really felt that knowing something and applying it are two different things… I did not know the exact protocol, so I just kind of kept pretending to have done it.*(Participant 2)

#### 3.3.2. School-Based Skills Training Not Reflecting Real-Life Clinical Settings

The participants stated that they could not practice HAI prevention and control in situations calling for it because they were too rushed. Although learning nursing skills in infection scenarios resembling clinical settings helps in performing infection prevention care, the students primarily underwent skills training in labs, as opposed to a scenario-based training, prior to their clinical practicum. Such training did not adequately prepare them to apply their skills in a clinical setting.
*What I should do when I am actually exposed to infection?… Ṭhe patient had an IV, and after removing the line, one has to apply pressure to stop the bleeding, but the patient’s bleeding had not completely stopped. So, when I went to take the patient’s blood pressure, the blood was still oozing. As I was in a hurry, I held onto that part with my bare hands, and called my nurse.*(Participant 1)
*I know, but when I actually do it, I sometimes miss out and forget about it… I think we should actually practice nursing skills, like trying to put on PPE and stuff … In a simulation, one is aware of the activities and performs them when given a situation… but in other labs, we are supposed to just visualize the situation when we practice… and I think that makes it harder to become aware of the activities.*(Participant 12)

### 3.4. Passive Learning Attitude

Although there were times when the preceptors prevented students from participating in nursing interventions for patients with infections, there were also times when the students themselves avoided entering these patients’ rooms or encountering them face-to-face. Additionally, lack of interest in infectious diseases and consequently not taking the time to read through the electronic medical records to gain relevant information were barriers to learning.

#### 3.4.1. Hesitancy about Learning about Patients with an Infection

After learning that a patient had an infectious disease, out of fear of contracting the infection the participants avoided entering isolation rooms and avoided situations that required them to provide nursing interventions to these patients. In particular, when students were exposed to patients who were confirmed to have an infection only after admission to the hospital, their anxiety was so severe that they could not focus during their rotations. This anxiety was a barrier to their learning about HAI prevention and control.
*At first, I was a little scared during the orientation. Would it be okay to go into this room? Would wearing only a gown and gloves be enough? So, at first, I was a little hesitant to go in… When I checked the vital signs of a patient with tuberculosis, I had so many concerns and I was a little hesitant to go in at that time. So, I think I tried to avoid the situation, rather than face it…*(Participant 2)
*I already knew that the patient had tuberculosis… so, I did not want to go in because I was scared.*(Participant 10)

#### 3.4.2. Lack of Interest

The participants had few opportunities to encounter patients with infections during their clinicals; thus, they had no interest in thinking about HAI prevention and control, or researching and learning about it. Students’ contact with patients with infections were further limited during clinical rotations since the outbreak of COVID-19; thus, they exhibited passive attitudes and did not bother to research the infection in the patients’ electronic medical records, even when they were allowed to do so.
*There were no instances when I had to think deeply about infection control; so, that may have been the reason why I did not attempt to search for things. If I had been exposed to such situations frequently, I would have tried to learn more about it, but that wasn’t the case. So, I think that is why I was not interested.*(Participant 2)
*I have come across CDI on electronic medical records, but I did not see those patients in detail; so, I did not intensively study infection control.*(Participant 7)

## 4. Discussion

Prevention and control of HAI has traditionally been an important parameter in Korea’s hospital accreditation evaluations for ensuring quality patient care, even prior to the COVID-19 pandemic. In the present study, we aimed to explore the barriers to learning HAI prevention and control among nursing students in Korea during clinical practicum. Our findings shed light on the barriers to learning HAI prevention and control to ameliorate limitations of the educational methods and environment, thereby ensuring that nursing students are taught HAI prevention and control practices that may be applied in all hospital environments.

Firstly, regardless of the pandemic status, the students experienced that they had limited opportunities to practice HAI control and prevention in clinical units. In Korea, students were not only prohibited from entering COVID-19 isolation rooms, but they also had minimal contact with patients with infections such as MRSA, tuberculosis, CDI, and HIV during their med-surg clinical rotations, as stated in previous studies [17,20]. Restricting students from contact with patients with infections compelled them to choose patients with non-communicable diseases as their patient of the day for their case study assignments, further reducing their learning opportunities for HAI control [20]. In particular, besides the disease-centered or theoretical learning approaches employed in a nursing educational structure [20], such environments for clinical practicum were a barrier to acquiring practice competencies among nursing students. As learning experiences in the hospital setting are an opportunity to affirm the knowledge learned in school and advance nursing skills [13,15,19], it is important to provide educational measures that allow nursing students to safely participate in nursing activities for patients with infections. Furthermore, nurse preceptors who oversee and guide students during clinical practicum did not strictly adhere to the best practices as role models, and this further challenged the students’ learning of HAI control practices. In a previous study [12] among nursing students in Ghana, the students stated that experiences, such as their clinical teachers not adhering to safety protocol or not supporting them, discouraged them from adhering to preventive care interventions. In particular, clinical preceptors’ failure to provide explanations about patients with infections to nursing students who are not experienced in recognizing them may prevent students from performing appropriate nursing interventions accordingly. A previous study on the experiences of nursing students in Poland who underwent clinical practicum during the COVID-19 pandemic [16], found that nurses who were uninspiring in their roles as clinical preceptors did not inform their students about patients’ COVID-19 status, which caused confusion among students in providing infection control. Therefore, clinical preceptors have a direct impact on nursing students’ practical HAI prevention and control competencies [10]. Hence, improving education for these nurses as preceptors and promoting their infection control activities will help provide students with more opportunities to learn about HAI prevention and control practices.

Secondly, the participants did not have detailed knowledge about precautions—standard, contact, droplet, or airborne—for HAI prevention and control. Hence, during the COVID-19 pandemic, other than hand hygiene, PPE usage, respiratory hygiene, or coughing etiquette, they lacked knowledge of which infection control activities to perform. In line with these findings, students have been reported to have a good foundation of knowledge on hand hygiene, PPE, and coughing etiquette, as part of standard precautions during the COVID-19 pandemic [12,14], but they still lack adequate knowledge about safe injection practices, sharps injury prevention, handling of laundry and linen, waste management, and decontamination of spilled waste [2,12]. Given that, nursing students’ practice of HAI prevention and control is influenced by their knowledge foundation [28], it is important to educate them about the specific components of standard precautions. Additionally, the loss of knowledge learned in the earlier years of nursing schools, by the time students begin clinical practicum in the third year [10,17], could be addressed by repeated education, such as refresher seminars every six months [28]. This would help them to retain their knowledge and, in turn, might drive their practical competencies.

Thirdly, the students faced challenges in performing nursing skills for HAI control because their skills were not refined and proficient when they began their clinical rotations. A previous study also reported that despite theoretically learning or practicing how to perform hand hygiene or don and doff PPE in a lab at school, students still had trouble performing these techniques during clinicals [12]. These infection control activities were reported to be learned through a trial-and-error process in clinical practice [15]. As online learning of infection control due to the COVID-19 pandemic poses challenges in providing safe patient care during clinical practicum, it is essential to provide adequate training before beginning clinical practicum [29]. In particular, our study’s students mentioned that they had limitations in recognizing infection-related situations and performing activities accordingly, especially when in a pressing situation. This was consistent with a previous report on nursing students in Ghana stating that emergency situations are barriers to infection prevention practices [12]. As students provide nursing interventions in a limited time window in an emergency, they forget safety precautions and infection prevention protocol. However, providing repeated education using realistic emergency or common scenarios could equip students with such competencies [17]. Simulation-based training [30], which offers the benefit of learning nursing skills in a scenario consisting of various case modules in a realistic and interactive setting, may be an effective option for bolstering students’ HAI prevention and control competencies.

Finally, as previous studies [15,16,18,19] have reported, participating in nursing activities for patients with infection induces fear among nursing students and such negative emotional responses served as a barrier to learning HAI prevention and control practices. Fear triggered by nursing activities for patients with infections stems from the uncertainty that they may not only contract the infection themselves but also spread it to others. This was the most distinct and intense emotion experienced before the development of COVID-19 vaccines, in the early days of the pandemic [16]. In addition, unwillingness to learn about patients with infections in isolation impedes students’ learning of practical competencies [10,20]. Therefore, instructors who educate nursing students about HAI prevention and control should gain an understanding of these emotional experiences and explore measures to support them to boost their learning competencies.

This study has a few limitations. It is an exploratory qualitative study; thus, its findings cannot be generalized to other regions and cultures. Nevertheless, its findings pertaining to significant barriers to learning HAI prevention and control practices in clinical practicum among nursing students will be useful for developing interventions to enhance their practical competencies. Another limitation is the small size of the focus groups. In 2019–2020, when this study was conducted, owing to the sudden outbreak of COVID-19, there was only one institution in our province where nursing students could practice. Our study is, therefore, meaningful in exploring the experiences of students who practiced in hospitals during the COVID-19 situation. Although we could not recruit large groups because of the COVID-19 pandemic, we did establish two separate groups for different school years and conducted several interview sessions to adequately convey the students’ experiences. Moreover, 12 participants’ data are commonly sufficient for homogenous sampling, such as in focus groups, whose participants have similar backgrounds and experience [31,32].

## 5. Conclusions

In this qualitative study, we used FGI to investigate the barriers to learning about HAI prevention and control among nursing students during clinical practicum. Its results revealed that nursing students have limited opportunities throughout their school curriculum and clinical rotations in hospitals to provide direct nursing care to patients with infections and ensure infection prevention, despite these being important targets for education. This restriction was found to hinder the expansion of knowledge that can be gained through face-to-face interactions with patients as well as the development of practical competencies and skills. Despite the COVID-19 pandemic and other factors, addressing these barriers in clinical practicum and safely involving nursing students in nursing activities for patients with infections is important because enhancing HAI prevention and control competences will adequately prepare these prospective nursing professionals who have to deal with infectious diseases throughout their careers.

## Figures and Tables

**Table 1 ijerph-20-06430-t001:** Key interview questions.

What was your experience of learning healthcare-associated infections prevention and control like during your clinical practicum?
What challenges have you faced when learning healthcare-associated infections prevention and control during your clinical practicum?
What things caused such challenges during your clinical practicum?
What were the barriers to learning during your clinical practicum?

**Table 2 ijerph-20-06430-t002:** Participants’ general characteristics (N = 12).

Participant Number	School Year	Sex (Age)	Courses in Which Students Recalled Having Learned Infection Control	Clinical Placement Units *
1	3	F (21)	Fundamentals of Nursing	CV, PED, OBGY, OS, OS
2	3	F (21)	Fundamentals of Nursing	INF, PED, OS, OBGY, OR
3	3	F (21)	Fundamentals of Nursing	CV, PED, OBGY, OS, OS
4	3	F (21)	Fundamentals of Nursing	INF, PED, OR, GS, OBGY
5	3	F (22)	Fundamentals of Nursing Adult Health Nursing	On, NICU, OS, OBGY, OR
6	3	M (23)	Fundamentals of Nursing	INF, PED, OR, OS, OBGY
7	4	M (25)	Fundamentals of Nursing	ICU, PED, GER, PSY, OS, On, NICU, ER, TS, OBGY
8	4	F (22)	Fundamentals of Nursing Microbiology	TICU, CNCU, GER, PSY, NR, PED, CV, OR, OS, OBGY
9	4	F (22)	Fundamentals of Nursing	GI, GER, TICU, PSY, CNCU, TS, OBGY, ER, PED, On
10	4	F (22)	Fundamentals of Nursing	PED, GER, ICU, PSY, PU, OBGY, OR, GS, NICU, On
11	4	F (23)	Fundamentals of Nursing	GER, ICU, PED, PSY, CNCU, IDM, NICU, OR, GS, OBGY
12	4	F (22)	Fundamentals of Nursing Adult Health Nursing	GER, ICU, PED, PSY, CNCU, OBGY, OR, OS, CV, NICU

* CNCU, comprehensive nursing care unit; CV, Cardiology; ER, emergency room; GER, Gerontology; GI, Gastroenterology; GS: general surgery; ICU, Intensive care unit; INF, Infectious Diseases Medicine; NICU, Neonatal intensive care unit; NR, neurology; OBGY, obstetrics & Gynecology; On, Hemato-oncology; OR, operating room; OS, orthopedic surgery; PED, Pediatrics; PSY, psychiatry; PU, pulmonology; TICU, Trauma intensive care unit; TS, Trauma surgery.

**Table 3 ijerph-20-06430-t003:** Categories and sub-categories identified in this study.

Categories	Sub-Categories	Frequencies *	Participant Number
Limited learning opportunities	Not many learning opportunities	4	4, 5, 11, 12
	Limited face-to-face interaction with patients with an infection	11	1, 2, 3, 4, 5, 7, 8, 9, 10, 11, 12
	Inability to be proactive in a clinical practicum setting	6	1, 2, 3, 5, 6, 11
Lack of knowledge about infection control	Lack of specific knowledge about transmission routes	12	1, 2, 3, 4, 5, 6, 7, 8, 9, 10, 11, 12
	Fading memories of learned information over time	5	1, 2, 3, 4, 8
Inadequate training experience	Inadequate infection management skills	7	2, 3, 4, 6, 8, 10, 11
	School-based skills training not reflecting real-life clinical setting	9	1, 2, 3, 4, 5, 6, 10, 11, 12
Passive learning attitude	Hesitant about learning about patients with an infection	4	1, 2, 7, 10
	Lack of interest	2	2, 4, 7

* Frequencies refer to total numbers of participants in each sub-category.

## Data Availability

Not applicable.

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
