# Peer review of "Barriers to Learning Healthcare-Associated Infections Prevention and Control during Clinical Practicum among Nursing Students in Korea: A Focus Group Study"

_ijerph, 2023, doi:10.3390/ijerph20146430_

Round 1
Reviewer 1 Report
Dear Authors,
I have read your manuscript with pleasure. I think the issue is relevant since preventing heathcare associated infections is very important in the hospital setting. As you aknowledge a limitation is the small number of nurses participating in the interviews, even taking into account that this is a qualitative study. Moreover, the fact that these nurses belong to a single center posses some doubt on the interpretation of the results. Barriers to education in HAI could be different among hospitals. Finally, I wonder if inclusion of some teaching nurses could have provided complementary data on the real situation in this hospital. Additionally, were there differences between second and third year students in their opinions?
Author Response
My co-authors and I, wish to resubmit our revised manuscript entitled “Barriers to Learning Healthcare-Associated Infection Prevention and Control during Clinical Practicum among Nursing Students in Korea: A Focus Group Study,” with changes that thoroughly address your comments.
We thank you for your thoughtful suggestions and insights. The manuscript has benefited tremendously from your thorough feedback. We look forward to working with you to move this manuscript closer to publication in IJERPH.
The manuscript has been rechecked, and the necessary changes have been made in accordance with your suggestions. The responses to all comments have been prepared and given below.
We appreciate your consideration.
Review 1 Comments: I have read your manuscript with pleasure. I think the issue is relevant since preventing healthcare associated infections is very important in the hospital setting:
Point 1: As you acknowledge a limitation is the small number of nurses participating in the interviews, even taking into account that this is a qualitative study. Moreover, the fact that these nurses belong to a single center posses some doubt on the interpretation of the results. Barriers to education in HAI could be different among hospitals.
Response 1: Thank you for your pertinent observation. Kuzel (1999) and Patton (2002) propounded that five to eight participants are sufficient in a qualitative study comprising homogeneous participants, such as a focus group, and 12–20 participants are appropriate when trying to achieve maximum variation. Therefore, for our study’s inquiry, we considered 12 participants as appropriate and request the reviewers’ consideration. We have also explained about the adequacy of the number of participants as a qualitative methodology in our focus group study in the Discussion section under limitations. (Lines 517-519)
Moreover, 12 participants’ data are commonly sufficient for homogenous sampling, such as in focus groups, whose participants have similar backgrounds and experience [31,32].
- Kuzel, A.J. Sampling in qualitative inquiry. In Doing Qualitative Research; Crabtree, B.F, Miller, W.L.,; Sage Publications: Thousand Oaks, CA, USA, 1999; pp. 33-45.
- Patton, M.Q. Designing qualitative studies [excerpt: Purposeful sampling]. In Qualitative Research and Evaluation Methods; ; Sage Publications: Thousand Oaks, CA, USA, 2002; pp.230-247.
Point 2: Moreover, the fact that these nurses belong to a single center posses some doubt on the interpretation of the results. Barriers to education in HAI could be different among hospitals.
Response 2: Thank you for your insightful comment. We have now explained the study’s limitations related to the prevailing situation. We hope you will consider that there is value in our study. (Lines 511-514)
In 2019–2020, when this study was conducted, owing to the sudden outbreak of COVID-19, our province had only one institution, where nursing students could practice. Our study is, therefore, meaningful in exploring the experiences of students, who practiced in hospitals during the COVID-19 situation.
Point 3: Finally, I wonder if inclusion of some teaching nurses could have provided complementary data on the real situation in this hospital.
Response 3: Thank you for your relevant query. We did not include nurse educators, as the students received practical training from nurses affiliated with university hospitals.
Point 4: Additionally, were there differences between second and third year students in their opinions?
Response 4: Thank you for your discerning question. As shown in Table 3 (Results section), participants representing the sub-categories were evenly distributed. Therefore, there was no difference in the barriers to learning about infection control, by grade. (Lines 163-165)

Reviewer 2 Report
Thank you so much for giving me the chance to review this remarkable paper, which is "Barriers to Learning Healthcare-Associated Infection Prevention and Control during Clinical Practicum among Nursing Students in Korea: A Focus Group Study." The authors apply an important topic to nursing students safety and protection measures against infections’ while working in healthcare facilities. This paper has been written and organised in a scientific way. I have some minor comments that I hope the authors consider. From my point of view, this manuscript should be published in IJERPH after the authors address my comments.
Abstract.
Based on my reading of the paper, the authors adopted a qualitative description study design to fulfil their study aims. So, please state the design clearly in the abstract section instead of using "qualitative study". P1, Line 16–17.
Introduction
As this study focuses on undergraduate nursing students, it is imperative to mention and highlight more about Korean nursing curricula, particularly the syllabus around infection control measures. Is there a separate course about infection and control measures? If yes, please mention this; if not, please try to estimate how many hours they attend during their undergraduate studies.
Method
In 2.2. Participants and Setting, the authors started with "Volunteering third-year and fourth-year nursing students in Korea….", I know what the authors mean by "volunteering," that the nursing students have a voluntary participation in the study. But some readers may understand that the students are volunteers in their third year. I suggest the authors revise the sentence or delete the word "volunteering.".
In the analysis section, the authors missed mentioning how they conducted the interview. Did they conduct the interview in Korean or English? If it was conducted in English, how did they ensure the comprehension and understanding of nursing for the interview guide? If it is conducted in Korean, how did the authors do the translation? The authors need to clearly mention the translation process?
Result and discussion
The result and discussion sections are well written.
Well done!
Author Response
Response to Reviewer 2 Comments
My co-authors and I, wish to resubmit our revised manuscript entitled “Barriers to Learning Healthcare-Associated Infection Prevention and Control during Clinical Practicum among Nursing Students in Korea: A Focus Group Study,” with changes that thoroughly address your comments.
We thank you for your thoughtful suggestions and insights. The manuscript has benefited tremendously from your thorough feedback. We look forward to working with you to move this manuscript closer to publication in IJERPH.
The manuscript has been rechecked and the necessary changes have been made in accordance with your suggestions. The responses to all your comments have been prepared and appended below.
We appreciate your consideration.
Review 2 Comments: Thank you so much for giving me the chance to review this remarkable paper, which is "Barriers to Learning Healthcare-Associated Infection Prevention and Control during Clinical Practicum among Nursing Students in Korea: A Focus Group Study." The authors apply an important topic to nursing students safety and protection measures against infections’ while working in healthcare facilities. This paper has been written and organised in a scientific way. I have some minor comments that I hope the authors consider. From my point of view, this manuscript should be published in IJERPH after the authors address my comments:
Point 1: Abstract. Based on my reading of the paper, the authors adopted a qualitative description study design to fulfil their study aims. So, please state the design clearly in the abstract section instead of using "qualitative study". P1, Line 16–17.
Response 1: Thank you for your valuable feedback. We have now stated the method used, namely, focus group interviews. In retrospect, our describing the research method later seems to have caused the confusion. (Line 17)
A qualitative study was performed using focus group interviews.
Point 2: Introduction. As this study focuses on undergraduate nursing students, it is imperative to mention and highlight more about Korean nursing curricula, particularly the syllabus around infection control measures. Is there a separate course about infection and control measures? If yes, please mention this; if not, please try to estimate how many hours they attend during their undergraduate studies.
Response 2: Thank you for your advice. We have now added about Korea’s nursing school curriculum, not including specialized courses or recommended programs for standardized HAI prevention and control. (Lines 54-56)
Korea’s nursing school curriculum comprises 1,000 or more clinical hours; yet, it does not include specialized courses or recommended programs for standardized HAI prevention and control [10-11].
Point 3: Method. In 2.2. Participants and Setting, the authors started with "Volunteering third-year and fourth-year nursing students in Korea….", I know what the authors mean by "volunteering," that the nursing students have a voluntary participation in the study. But some readers may understand that the students are volunteers in their third year. I suggest the authors revise the sentence or delete the word "volunteering."
Response 3: Thank you for your kind recommendation. We have now rephrased the sentence. (Lines 81-82)
Third and fourth-year nursing students in Korea, with clinical practicum experience in a hospital setting, voluntarily participated in this study.
Point 4: Method. In the analysis section, the authors missed mentioning how they conducted the interview. Did they conduct the interview in Korean or English? If it was conducted in English, how did they ensure the comprehension and understanding of nursing for the interview guide? If it is conducted in Korean, how did the authors do the translation? The authors need to clearly mention the translation process?
Response 4: We are grateful for your thorough review. We conducted the research in Korean. Before submitting the study’s results to IJERPH, the researchers had it translated into English, and edited. We have added this description in the analysis section. (Lines 132-134)
Each process of this study was performed in Korean. For publication, this manuscript was translated in English by professional translator and edited by expert.
Point 5: Result and discussion. The result and discussion sections are well written. Well done!
Response 5: Thank you for your appreciation, which has truly motivated us.

Reviewer 3 Report
In the current article (ijerph-2403090), the authors highlighted the different barriers faced by the nursing students in Korea while learning prevention and control of Hospital Acquired infections. The findings of the study will be helpful for Nursing students to develop effectively acquire infection control competencies during clinical work. The following comments should be properly addressed.
Comments:
- In the title of the article, Healthcare-Associated Infection may be changed to Healthcare-Associated Infections. Rationale of the study should be clearly written.
- The study is performed in which hospital of Korea?
- What was the control group of the study?
- The statistical analysis should be performed to make the data more reliable.
- The number of participants (10) are very small. Please justify???
- Four themes and eight categories were selected for collecting data. Scope seem very narrow, please justify???
- In table 3, data may be arrange in more quantitative manner (such as percentages).
- Why only two male subjects were recruited in the study?
- The limitation of the study should be properly described.
There are some grammatical and formatting mistakes which should be removed.
Author Response
Response to Reviewer 3 Comments
My co-authors and I, wish to resubmit our revised manuscript entitled “Barriers to Learning Healthcare-Associated Infection Prevention and Control during Clinical Practicum among Nursing Students in Korea: A Focus Group Study,” with changes that thoroughly address your comments.
We thank you for your thoughtful suggestions and insights. The manuscript has benefited tremendously from your thorough feedback. We look forward to working with you to move this manuscript closer to publication in IJERPH.
The manuscript has been rechecked, and the necessary changes have been made in accordance with your suggestions. Responses to all the comments have been prepared and given below.
We appreciate your consideration.
Review 3 Comments: In the current article (ijerph-2403090), the authors highlighted the different barriers faced by the nursing students in Korea while learning prevention and control of Hospital Acquired infections. The findings of the study will be helpful for Nursing students to develop effectively acquire infection control competencies during clinical work. The following comments should be properly addressed:
Point 1: In the title of the article, Healthcare-Associated Infection may be changed to Healthcare-Associated Infections. Rationale of the study should be clearly written.
Response 1: Thank you for your kind recommendation. We have now revised the title. (Lines 2-4)
Barriers to Learning Healthcare-Associated Infections
Prevention and Control during Clinical Practicum among Nursing Students in Korea: A Focus Group Study
Point 2: The study is performed in which hospital of Korea?
Response 2: Thank you for your advice. We have now added the location of the hospital in the Methods section. (Lines 90-92).
The site for the clinical practicums was a national university general hospital in Korea, Cheongju. This hospital was nationally designated for isolation beds during the COVID-19 pandemic.
Point 3: What was the control group of the study?
Response 3: Thank you for your careful review and valid query. This study did not have a specific control group. This is because its research methodology, comprised a focus group, in which a researcher generates an opinion on a specific topic from participants who have similar experiences. So, this study commonly collected data through interactions with a research group. We request your kind consideration for our research technique.
Point 4: The statistical analysis should be performed to make the data more reliable.
Response 4: Thank you for your valuable comments. Unlike quantitative content analysis, qualitative content analysis generally does not use statistical analysis. While quantitative content analysis expresses statistic data, qualitative content analysis derives categories after careful examination of original data, such as transcripts of interview-recordings. As this study’s data were analyzed using qualitative content analysis, we made our own decisions about setting categories. We have elaborated on this study’s analysis procedures in the Materials and Methods section. The relevant reference is listed below.
Bengtsson, M. How to plan and perform a qualitative study using content analysis. NursingPlus Open 2016, 2, 8-14. https://dx.doi.org/10.1016/j.npls.2016.01.001
“Content analysis is unique in that it has both a quantitative and a qualitative methodology. In quantitative content analysis, facts from the text are presented in the form of frequency expressed as a percentage or actual numbers of key categories. This method summarizes rather than reports all details concerning a message set, and the researcher seeks to answer questions about how many. In qualitative content analysis, data are presented in words and themes, which makes it possible to draw some interpretation of the results.”
Point 5: The number of participants (10) are very small. Please justify?
Response 5: Thank you for your important comment. According to Kuzel (1999) and Patton (2002), 5–8 participants are sufficient in a qualitative study for homogeneous participants, such as a focus group, and 12–20 participants are considered appropriate when trying to achieve maximum variation. Moreover, Guest, Bunce, and Johnson (2006) averred that data saturation generally occurs within the first 12 interviews. Therefore, for our study’s inquiry, we considered 12 participants appropriate. We request the reviewers to kindly consider our viewpoint. We have explained about the adequacy of the number of participants in our focus group study as a qualitative methodology in the Discussion section under limitations. (Lines 517-519).
Moreover, 12 participants’ data are commonly sufficient for homogenous sampling, such as in focus groups, whose participants have similar backgrounds and experience [31,32].
- Kuzel, A.J. Sampling in qualitative inquiry. In Doing Qualitative Research; Crabtree, B.F, Miller, W.L.,; Sage Publications: Thousand Oaks, CA, USA, 1999; pp. 33-45.
- Patton, M.Q. Designing qualitative studies [excerpt: Purposeful sampling]. In Qualitative Research and Evaluation Methods; ; Sage Publications: Thousand Oaks, CA, USA, 2002; pp.230-247.
Guest, G.; Bunce, A.; Johnson, L. How many interviews are enough? An experiment with data saturation and variability. Field Methods. 2006, 18, 59–82; Doi:10.1177/1525822X05279903
(p 59) “Based on the data set, they found that saturation occurred within the first twelve interviews, although basic elements for meta-themes were present as early as six interviews.”
(p 79) “Six to twelve interviews will always be enough to achieve a desired research objective, or using the findings above to justify research… For most research enterprises, in which the aim is to understand common perceptions and experiences among a group of relatively homogeneous individuals, twelve interviews should suffice.”
Point 6: Four themes and eight categories were selected for collecting data. Scope seem very narrow, please justify?
Response 6: We are grateful for your thorough review. We classified the data into four categories and eight sub-categories. Since our study conducted content analysis using the general inductive qualitative data analysis method, we believe that our number of categories were appropriate, based on the composition of three to eight categories proposed by Tomson (2006). We request the reviewers to kindly consider our standpoint. The relevant reference is listed below.
Tomas, D.R. A general inductive approach for analyzing qualitative evaluation data. Am. J. Eval. 2006, 237-246; DOI:10.1177/109821400528374
(p242) “The intended outcome of the process is to create a small number of summary categories (e.g., between three and eight categories) that in the evaluator’s view capture the key aspects of the themes identified in the raw data, and are assessed to be the most important themes given the evaluation objectives.”
Point 7: In table 3, data may be arrange in more quantitative manner (such as percentages).
Response 7: Thank you for your kind recommendation. As clarified in point 4, unlike quantitative content analysis, qualitative content analysis generally does not use statistical analysis. While quantitative content analysis expresses statistical data, qualitative content analysis derives categories after careful examination of original data, such as transcripts of interview-recordings. As a supplementary method, we have compiled Table 3 to represent the participants’ observations on the derived sub-categories. We request you to please consider our reasoning for choosing not to present in a statistical manner, the derived categories and sub-categories. The relevant reference is listed below.
Bengtsson, M. How to plan and perform a qualitative study using content analysis. NursingPlus Open 2016, 2, 8-14. https://dx.doi.org/10.1016/j.npls.2016.01.001
“Content analysis is unique in that it has both a quantitative and a qualitative methodology. In quantitative content analysis, facts from the text are presented in the form of frequency expressed as a percentage or actual numbers of key categories. This method summarizes rather than reports all details concerning a message set, and the researcher seeks to answer questions about how many. In qualitative content analysis, data are presented in words and themes, which makes it possible to draw some interpretation of the results.”
Point 8: Why only two male subjects were recruited in the study?
Response 8: Thank you for your careful review and valid observation. Only two male students expressed their willingness to participate voluntarily through purposeful sampling. The proportion of male students in the target grade for participant sampling was about 10%, but as this study’s purpose was not to identify gender differences, the distribution of participants by gender was not a consideration in the research method.
Point 9: The limitation of the study should be properly described.
Response 9: Thank you for your kind recommendation. We have now added descriptions about the sample size and institution, under limitations (Lines 510-519).
Another limitation is the small size of the focus groups. In 2019–2020, when this study was conducted, owing to the sudden outbreak of COVID-19, there was only one institution in our province where nursing students could practice. Our study is, therefore, meaningful in exploring the experiences of students, who practiced in hospitals during the COVID-19 situation. Although we could not recruit large groups because of the COVID-19 pandemic, we did establish two separate groups for different school years, and conducted several interview sessions to adequately convey the students’ experiences. Moreover, 12 participants’ data are commonly sufficient for homogenous sampling, such as in focus groups, whose participants have similar backgrounds and experience [31,32].
Point 10: Comments on the Quality of English Language. There are some grammatical and formatting mistakes which should be removed.
Response 10: Based on your advice, we have carefully reviewed the grammar and edited the text.

Reviewer 4 Report
Thank you very much for inviting me to prepare a review of the manuscript “Barriers to Learning Healthcare-Associated Infection Prevention and Control during Clinical Practicum among Nursing Students in Korea: A Focus Group Study”
In my opinion, all parts of the manuscript are presented very accurately and correctly.
Qualitative research - well designed Introduction - the review of the literature was properly conducted, introducing the subject and justifying the choice of research. Well-described qualitative method - focus groups and participants and setting and data collection The results are clear, divided into subtopics – correct. Discussion – correct. The authors listed the limitations of the study with which I agree. The conclusions - they give practical tips.
Author Response
Response to Reviewer 4 Comments
My co-authors and I, wish to resubmit our revised manuscript entitled “Barriers to Learning Healthcare-Associated Infection Prevention and Control during Clinical Practicum among Nursing Students in Korea: A Focus Group Study” with changes that thoroughly address your comments.
We thank you for your thoughtful suggestions and insights. The manuscript has benefited tremendously from your thorough feedback. We look forward to working with you to move this manuscript closer to publication in IJERPH.
The manuscript has been rechecked, and the necessary changes have been made in accordance with your suggestions. Responses to all the comments have been prepared and given below.
We appreciate your consideration.
Review 4 Comments: Thank you very much for inviting me to prepare a review of the manuscript “Barriers to Learning Healthcare-Associated Infection Prevention and Control during Clinical Practicum among Nursing Students in Korea: A Focus Group Study”.
In my opinion, all parts of the manuscript are presented very accurately and correctly:
Qualitative research - well designed Introduction - the review of the literature was properly conducted, introducing the subject and justifying the choice of research.
Well-described qualitative method - focus groups and participants and setting and data collection.
The results are clear, divided into subtopics – correct.
Discussion – correct. The authors listed the limitations of the study with which I agree.
The conclusions - they give practical tips.
Thank you your kind and careful review. We are truly encouraged by your positive feedback.
